# Isolation and Characterization of Extracellular Vesicles Derived from Mango Fruits

**DOI:** 10.3390/ijms262311375

**Published:** 2025-11-25

**Authors:** Aleksandra Steć, Grzegorz Szaknis, Anna Skowrońska, Przemysław Mielczarek, Grzegorz S. Czyrski, Luna Gade, Andrea Heinz, Agata Płoska, Leszek Kalinowski, Bartosz Wielgomas, Szymon Dziomba

**Affiliations:** 1Department of Toxicology, Faculty of Pharmacy, Medical University of Gdansk, 107 Hallera Street, 80-416 Gdansk, Poland; aleksandra.stec@gumed.edu.pl (A.S.); bartosz.wielgomas@gumed.edu.pl (B.W.); 2Department of Analytical Chemistry and Biochemistry, Faculty of Materials Science and Ceramics, AGH University of Science and Technology, 30 Mickiewicza Avenue, 30-059 Krakow, Poland; przemyslaw.mielczarek@agh.edu.pl; 3Laboratory of Proteomics and Mass Spectrometry, Maj Institute of Pharmacology, Polish Academy of Sciences, 12 Smetna Street, 31-343 Krakow, Poland; 4LEO Foundation Center for Cutaneous Drug Delivery, Department of Pharmacy, University of Copenhagen, 2100 Copenhagen, Denmark; grzegorz.czyrski@sund.ku.dk (G.S.C.); lunagade@gmail.com (L.G.); andrea.heinz@sund.ku.dk (A.H.); 5Department of Medical Laboratory Diagnostics—Fahrenheit Biobank BBMRI.pl, Faculty of Pharmacy, Medical University of Gdansk, 7 Debinki Street, 80-211 Gdansk, Poland; agata.ploska@gumed.edu.pl (A.P.); leszek.kalinowski@gumed.edu.pl (L.K.); 6BioTechMed Centre, Department of Mechanics of Materials and Structures, Gdansk University of Technology, 11/12 Narutowicza Street, 80-233 Gdansk, Poland

**Keywords:** capillary electrophoresis, exosomes, Mangifera, markers, plant, proteomics, size-exclusion chromatography

## Abstract

The mango (*Mangifera indica* L.) is a commonly cultivated tropical fruit across the globe. It is known to be rich in carotenoids, polyphenols, and vitamins, compounds that largely account for its nutritional and medicinal properties. Although the beneficial effects of mango phytochemicals have been widely documented, virtually no studies have investigated extracellular vesicles (EVs) originating from mango fruit. In the presented work, we developed a workflow combining differential centrifugation, filtration, and size-exclusion chromatography for the isolation of EVs from mango pulp. The isolates were characterized in accordance with the guidelines of the International Society of Extracellular Vesicles recommendations. The optimized size-exclusion chromatography column, packed with Sepharose CL-6B beads, enabled the recovery of a high-quality EV fraction, which was characterized in terms of physicochemical properties. Additionally, proteomic analysis identified 1084 proteins, many of which are associated with antioxidant, antimicrobial, and anti-inflammatory functions. These findings provide the first comprehensive characterization of mango-derived EVs and suggest that they may contribute to the biological activity traditionally attributed to mango consumption.

## 1. Introduction

Extracellular vesicles (EVs) are a heterogeneous group of submicron-sized structures secreted by living cells [1,2]. EVs play an important role in the physiology of organisms as well as in the pathogenesis of various diseases. Due to their ability to transport biomolecules like metabolites, nucleic acids, and proteins, EVs take part in the interplay between cells of the same organism and between species [1,2,3].

In plants, EVs are increasingly recognized as key mediators of intercellular communication. Plant EVs have been shown to participate in stress responses, immune signaling, and interactions with microorganisms, including both pathogenic and symbiotic species [1,3]. Plant-derived EVs are generally considered non-toxic to human cells [4,5,6] and are recognized as an important alternative source of EVs [7]. In comparison to mammalian specimens, plant material is readily available, inexpensive, and sustainable. Several plant sources have been explored for EV isolation, including edible fruits and vegetables [4,5,8,9,10,11,12]. Antioxidant, anti-inflammatory, antimicrobial, and tissue regenerative properties are among those frequently reported [2]. Such activities are attributed to the diverse molecular cargo of plant-derived EVs, which includes secondary metabolites (e.g., flavonoids, terpenes, and anthraquinones) [13,14], proteins [15,16], and mRNA [17]. These components can interact with cellular signaling pathways, catalyze specific biochemical processes, and modulate gene and protein expression in recipient cells. Moreover, several studies have demonstrated the successful delivery of xenobiotics (like chemotherapeutics, proteins, and nucleic acids) to mammalian cells both in vitro and in vivo using plant-derived EVs as natural carriers [6,18,19]. These findings indicate that plant-derived vesicles could represent a versatile platform for drug delivery and nutraceutical applications.

Mango (*Mangifera indica* L.) is one of the most widely cultivated tropical fruits globally. Besides its economic and agricultural significance, the mango holds a prominent place in traditional medicine [20]. The fruits are a rich source of vitamins (C, A, and B group), polyphenols (mangiferin, quercetin, and kaempferol), and fiber [20,21]. Extracts from mango have been shown to feature antioxidant properties and inhibit tumor growth [22]. Although the therapeutic value of mango’s phytochemical composition is well established, no attention has been paid yet to mango-derived EVs, which may be related to technical challenges inherent to working with mango juice and pulp. These matrices are rich in pectins and other cell wall polysaccharides, which are prone to co-precipitating with the vesicles, forming gel-like structures that interfere with common isolation techniques.

In this work, we developed a methodology for the isolation of EVs from the juice of mango fruits. For this purpose, size-exclusion chromatography (SEC) columns were developed. The elution profile of the columns was assessed with various analytical techniques to select the most appropriate material for EV characterization. Finally, nanoparticle tracking analysis (NTA), cryogenic transmission electron microscopy (cryo-TEM), electrophoretic light scattering (ELS), and mass spectrometry (MS) were employed for mango EVs’ analysis. The findings provide a basis for further exploration of mango EVs’ health-promoting properties.

## 2. Results

### 2.1. SEC Column Characterization

The fractionation of plant material was performed in columns packed with 3 mL of Sepharose CL-6B bead suspension. The isolation procedure is described in Section 4.2. The obtained fractions were characterized with NTA, BCA, and CE analyses. According to the NTA, more than 80% of particles were eluted in the third and fourth fractions (Figure 1A). Although both these fractions were rich in proteins (Figure 1B), the purity of third fraction was superior considering the particle-to-protein (PtP) criterion (Figure 1C) [23].

These observations were confirmed with CE. The analyses showed no detectable components in the first two fractions (Figure 2A). A signal typical for EVs was detected in the third and fourth fractions. The complexity of the fifth–eighth fractions was too high to confirm the presence of the EV signal. Isolates staining with SYBR Gold reagent and CE-LIF analysis (Figure 2B) confirmed the identity of the signal and revealed its presence in fractions 3–6. Interestingly, another fluorescent component (marked with an asterisk) was detected in fractions 4–7. However, this compound was not detected in the third fraction. Moreover, the third fraction was found to be devoid of virtually any co-isolates (Figure 2A). This observation was in line with the PtP ratio results (Figure 1C). Due to the superior purity, the third fraction obtained during SEC separation was selected for further characterization.

### 2.2. Physical Characterization of the Mango-Derived EVs

NTA measurements of the third fraction revealed the presence of nanoparticles with a diameter ranging up to 160 nm (Figure 3A). The mean, mode, and median sizes of the vesicles were 72 ± 1 nm, 65 ± 4 nm, and 69 ± 2 nm, respectively (five measurements of three independently prepared isolates). The total yield obtained in a single separation, estimated based on the NTA and BCA measurements, was 1.3 × 10^11^ ± 1.7 × 10^10^ particles (five measurements of three independently prepared isolates) and 59 ± 7 µg of proteins (two measurements of three independently prepared isolates).

Cryo-TEM analysis confirmed the presence of round-shaped structures with clearly distinguishable membranes (Figure 3B and Appendix A). The microscopic images indicated low EV concentration, which was in line with the quantitative measurements performed with BCA and NTA. The vast majority of detected particles were <100 nm, which is in good agreement with NTA results.

ELS analysis showed a homogeneous charge distribution among the EV population (Appendix A). The zeta potential was −15.9 ± 0.6 mV, which is lower than the zeta potential values typically reported for plant-derived EVs (>20 mV) [6,24].

### 2.3. Proteomic Analysis of the Mango-Derived EVs

The proteomic analysis allowed us to identify 1084 unique proteins in the isolates (Appendix A). Of note, 195 proteins were identified in all three replicates. Enzymes taking part in cell metabolism (glycolytic pathway, citric acid cycle, and others) and intercellular signaling (14-3-3, kinases, and annexin/calmodulin-type proteins, etc.) were the most abundant. Numerous proteins were linked to vesiculation, inner organelles, and cellular plasma membrane, including Rab GTPases, clathrin coat proteins, aquaporins, tetraspanin and syntaxin family members, and patellins. Chaperone proteins (HSP70 family members, luminal-binding protein, etc.) and proteins relevant to the biosynthesis and modification of mango secondary metabolites (flavonoids, mangiferin derivatives, terpenoids) were also identified. Antioxidant and reactive oxygen species (ROS)-related proteins (germin-like proteins, peroxidases, superoxide dismutase, and other redox enzymes), as well as antimicrobial proteins (chitinases, β-1,3-glucanases, and PR proteins, etc.), detected in this study, indicate the potential biological activity and function of mango-derived EVs.

Among the proteins related to vesiculation, the presence of TET-8 was confirmed. This particular tetraspanin is currently considered one of the very few established markers of plant-derived vesicles formed in the endosomal biosynthesis pathway (so-called “plant-derived exosomes”) [25]. Interestingly, the second postulated marker, TET-9 protein, was not detected in the presented study. The presence of syntaxin-121 (PEN1) was also confirmed, which is often linked with the presence of plant microvesicles [25]. The presence of other recently recommended plant-derived EV proteomic markers was also confirmed, including aquaporins, V-type proton ATPase subunits, clathrin chains, germin-like proteins, and fasciclin-like arabinogalactan protein (isoforms 1, 7, 8, and 13) [26]. Especially the isoforms 10 and 13 of the latter protein have been postulated to be specific for plant-derived EVs. Other, less specific plant-derived EV markers were also detected (HSP70, 14-3-3 proteins, and patellins).

Among the potential EV markers, calreticulin and Exo70E2 were not detected. The latter protein is an exocyst-forming subunit that has been proposed as a marker of an alternative vesicle secretory pathway in plant cells [27]. However, other exocyst complex components were identified, like Exo70A1 protein, which is also often reported in plant-derived EV isolates [28,29].

## 3. Discussion

Mango (*Mangifera indica* L.) is one of the most widely cultivated tropical fruits worldwide. Beyond its economic relevance, mango is highly valued for its nutritional and therapeutic potential, being a rich source of polyphenols, carotenoids, vitamins, and the bioactive xanthone mangiferin [20,21,30]. These compounds have been extensively studied for their antioxidant, anti-inflammatory, antimicrobial, antidiabetic, and anticancer properties, which support the fruit’s relevance not only in human nutrition but also in preventive and complementary medicine. Considering the well-documented bioactivity of mango metabolites, EVs derived from this fruit may provide unique biological activities and health benefits.

The developed methodology for EV isolation includes filtration, differential centrifugation, and SEC. The first two techniques enabled the removal of solid elements of the fruit pulp, cells, cell debris, and other submicron-sized particles. According to NTA measurements, no particles with a diameter above 160 nm were observed (Figure 3A). The purification of the vesicles was performed with a developed SEC column. The superior purity of the third fraction over the others was demonstrated with the PtP ratio parameter and with CE analysis. Only this fraction was selected for further experiments (cryo-TEM and proteomic analysis). In spite of relatively poor process recovery (third fraction contained 39.8 ± 5.1% of the total particle content) and isolation yield (59 ± 7 µg of proteins), the methodology was found to be sufficient for the proteomic analysis of mango-derived EVs. Furthermore, due to the scalability of the chromatography, the extension of the column length might be considered in future experiments if necessary.

The relatively low yield of a single isolation process can be attributed to the lack of preconcentration of the sample before the purification process. The attempts made to enrich the sample with centrifugal concentration (Vivaspin 20, 100 kDa pore size, PES membrane, Cytiva) resulted in rapid sample gelation and filter clogging. As a result, only 0.5 mL of raw mango juice was processed during a single chromatographic separation. However, it was possible to combine EV-rich fractions, obtained in multiple SEC runs, and to reduce the isolate volume with ultrafiltration (according to the description in Section 4.2) to achieve greater EV concentration, which confirms the efficient pectin removal during SEC separation.

Proteomic analysis of EVs isolated from mango juice revealed a complex protein composition. Enzymes associated with the central metabolism, vesiculation, and signaling constituted a significant part of the identified proteins. Such proteins are commonly found in plant-derived EVs and are assumed to reflect the metabolic and structural background of their parent cells rather than unique, cargo-specific functions [31]. Their presence reflects the processes essential for EV biogenesis, stability, and transport. Similar metabolic and vesicle-associated proteins, including glycolytic enzymes, Rab GTPases, SNAREs, and clathrin components, have been consistently reported across diverse plant species [26,28,31], supporting the conserved mechanisms underlying EV formation and release.

The proteomic profile of mango-derived EVs included numerous enzymes linked to the biosynthesis of secondary metabolites. Proteins such as polyphenol oxidases, glycosyltransferases, and shikimate pathway enzymes are known to participate in the synthesis of bioactive molecules like mangiferin, quercetin, and kaempferol, responsible for the health-promoting properties of mango [20,21]. The detection of such biosynthetic enzymes in the EV isolates suggests that these vesicles may be derived from metabolically active cells or be selectively incorporated with biosynthetic machinery during their formation. This observation leads to the hypothesis that mango EVs might also carry low-molecular-weight phytochemicals, which have already been reported in other plant species [13,24].

Enzymes relevant for the regulation of redox homeostasis in plant cells were identified in the EV isolates. Their presence might indicate antioxidant activity, which is frequently reported for plant-derived EVs [32]. The antioxidative properties of mango-derived EVs might be linked to the presence of polyphenols in the vesicle cargo. The antioxidative and cytoprotective properties of plant-derived EVs obtained from various species have recently been attributed to their secondary metabolite activities [5,13,33,34]. Several studies have demonstrated that plant-derived EVs can attenuate oxidative stress and inflammatory responses in mammalian systems by influencing AhR/Nrf2 and ERK1-2/NF-κB signaling pathways [13,15,33,35], which was also linked to the phenolic composition of the vesicles [13,33]. Regardless of the mechanism of action, which is most likely complex and involves various components of the EVs, the antioxidant activity of mango-derived EVs remains probable. This mechanism of action of plant-derived EVs has already been demonstrated to be effective in ROS scavenging, wound healing, tumor growth inhibition, and skin senescence reduction [32]. Considering that mango juice and fruit extracts exhibit strong antioxidant, anti-inflammatory, and cytoprotective properties [20,21], these effects may be partially mediated by EVs naturally present in mango tissue. Such a hypothesis links the beneficial properties of mango to a vesicle-based delivery of enzymatic and phenolic antioxidants.

A considerable fraction of the identified proteins in mango-derived EVs were molecular chaperones, including HSPs and other chaperonin family members. Although these proteins are not specific to plant-derived EVs [26] and their role as plant vesicle cargo components has not been explored, their presence reflects the involvement of stress-related pathways in EV biogenesis. In mammals, EV-carried HSPs have been reported to mediate intercellular communication within the tumor microenvironment [36]. Moreover, HSPs associated with EVs have been shown to act as signaling molecules that modulate immune responses and influence cell survival [37]. Thus, the abundance of chaperone proteins in mango EVs may not only indicate their role in maintaining vesicle integrity under stress conditions but also suggest potential biological activity upon interaction with mammalian cells.

Several identified proteins are associated with plant defense mechanisms, including pathogenesis-related proteins, chitinases, β-1,3-glucanases, and germin-like proteins. These proteins are key components of the plant immune system and are known to exhibit antibacterial, antifungal, and antiviral activities [38]. While the involvement of EVs in the response to infection is well-documented in plants [39], the presence of antimicrobial proteins indicates a similar role of EVs also in the case of mango.

Although this study provides the first comprehensive proteomic characterization of EVs derived from mango fruit, several limitations should be acknowledged. The conclusions drawn regarding the potential biological activities of isolated vesicles are based solely on proteomic data. They may vary depending on the cultivation region, plant variety, growth conditions and potential stress conditions. These factors were not controlled in the presented work which should be taken into consideration for data interpretation. While the identified protein cargo suggests multiple possible functional pathways and bioactivities, other important molecular components of EVs were not analyzed. Low-molecular-weight metabolites, lipids, and nucleic acids are expected to participate in EV-mediated signaling and biological regulation, and their absence in the present analysis limits the scope of functional interpretation.

Future studies should therefore aim to complement the proteomic findings with metabolomic and transcriptomic (mRNA and small RNA) analyses to provide a more complete understanding of the molecular composition of mango EVs. Further work should also include evaluation of EV stability after consumption. Although high resistance of EVs to gastrointestinal digestion has been reported in a few publications [40,41,42], several other articles demonstrated at least partial degradation of EVs with limited deterioration of vesicle cargo and activity [43,44,45,46]. This is why functional assays using not only mammalian cell models but also animal systems are essential to confirm the biological relevance of these vesicles and to validate the potential activities resulting from the proteomic data. Such integrative approaches will help to elucidate the mechanisms through which mango-derived EVs may exert physiological effects and expand their potential applications in nutrition, pharmacology, and nanobiotechnology.

## 4. Materials and Methods

### 4.1. Chemicals

Bovine serum albumin (BSA), disodium tetraborate decahydrate, phosphate buffered saline (PBS), and sodium dodecyl sulfate (SDS), were obtained from Merck (Steinheim, Germany). Sodium hydroxide pellets were purchased from J.T. Baker Chemicals (Center Valley, PA, USA). All chemicals were of analytical grade. In experiments, deionized water (resistivity of 18.2 MΩ cm) was used.

### 4.2. Isolation of Plant EVs

Mango fruits (varieties Kent, Haden or Osteen—depending on the availability) were obtained from a local grocery shop. Fruits were washed with lukewarm water, peeled, and mechanically squeezed. The juice was centrifuged for 30 min at low speed (3000× *g*; 4 °C) in a Sorvall 16R centrifuge (Thermo Fisher Scientific, Waltham, MA, USA) using a swing-out bucket. The supernatant was collected and centrifuged using greater g-forces (10,000× *g*) using a fixed-angle rotor (30 min, 4 °C). The obtained supernatant was filtered using polyethersulfone (PES) syringe filters: 5 µm (Chromafil Xtra; Macherey-Nagel; Duren; Germany) and 0.22 µm (Filtropur S 0.2; Sarstedt; Göttingen Germany). The filtrate was then centrifuged using 25,000× *g* (30 min, 4 °C). The supernatant (0.5 mL) was fractionated with a Sepharose CL-6B (Cytiva, Marlborough, MA, USA) cartridge.

Sepharose CL-6B beads were packed in 4.5 mL polypropylene cartridges. In each cartridge, 3 mL of the Sepharose bead suspension was packed using gravimetric flow and 25 mM borate buffer (pH 9.2) solution as an eluent. A minimum of six fractions of 0.4 mL each were collected for the analysis. For downstream analyses (microscopy and proteomic analysis), the third fractions obtained in multiple SEC separations (typically four) were combined and preconcentrated with a centrifugal concentrator (Amicon Ultra 0.5; 50 kDa, PES membrane, Merck, Steinheim, Germany) down to 0.5 mL volume. Next, G-25 columns (Cytiva, Marlborough, MA, USA) were used to replace the buffer with PBS. The collected fractions (second and third) were combined and preconcentrated with Amicon Ultra 0.5 concentrator down to 150 µL.

### 4.3. Bicinchoninic Acid Assay (BCA)

Protein content was measured using the Pierce BCA kit (Thermo Fisher Scientific, Waltham, MA, USA) according to the vendor’s recommendations. Concentrations were determined based on the calibration curve constructed with BSA standard solutions. Both samples and standards were mixed with 6% SDS solution in a 9 to 1 volumetric ratio. Next, 10 µL of such a solution was transferred to a 96-well plate and mixed with 200 µL of BCA reagents mixture. The plate was incubated at 37 °C for 30 min. The absorbance was measured using an Infinite 200 plate reader (Tecan, Mannedorf, Switzerland) at 562 nm.

### 4.4. Nanoparticles Tracking Analysis (NTA)

NTA measurements were performed using a Nanosight NS300 instrument (Malvern Instruments, Worcestershire, UK) with NTA software (version 3.2 Dev Build 3.2.16, Malvern Instruments, Worcestershire, UK). The required concentration range (10^7^–10^9^ particles mL^−1^) was obtained with a 25 mM borate buffer (pH 9.2) solution unless otherwise stated. Measurements were done in duplicate using a 405 nm laser and an sCMOS camera. At least 3 videos (1 min each) were recorded. The camera level was set to 15 or 16, with a focus ranging between 180 and 220 at 25 °C.

### 4.5. Capillary Electrophoresis (CE)

CE experiments were performed with a PACE MDQ plus system (Sciex, Framingham, MA, USA). Absorbance and fluorescence were measured with a diode array detector (DAD) and a laser-induced fluorescence (LIF) detector, respectively. When the DAD detector was used, the analyses were monitored at 200 nm. In the case of LIF detection, a 488 nm laser was used for excitation, and the emission was measured at 520 nm.

Electrophoresis was conducted using uncoated fused silica capillaries (50 µm i.d. × 363 µm o.d. × 30.2 cm of total length) at a positive voltage of 10 kV and constant temperature (25 °C). The background electrolyte (BGE) was composed of 25 mM borate buffer (pH 9.2). Samples were injected hydrodynamically (5 s, 3.45 kPa). The capillary was conditioned at the beginning of each working day with 0.1 M NaOH solution, water, and BGE for 10 min each at 137.9 kPa. Before every analysis, the capillary was rinsed with 0.1 M NaOH solution (2 min), water (1 min), and BGE (2 min). A water dipping procedure was performed before sample injection, and a short post-injection plug of BGE was used (5 s, 3.45 kPa).

The samples were analyzed directly (DAD detector) or after staining (LIF detection). SYBR Gold (Invitrogen, Waltham, MA, USA) dye 10,000× concentrate was diluted 100-fold with DMSO and mixed in a 1:9 volumetric ratio with the samples. After 1 h incubation at room temperature and in the dark, the samples were analyzed with CE without any further processing.

### 4.6. Cryogenic Transmission Electron Microscopy (Cryo-TEM)

The morphology of the isolated EVs was investigated by cryo-TEM. Prior to visualization, Lacey F/SiO 300 mesh Cu grids (Ted Pella, Inc., Redding, CA, USA) were glow-discharged at 10 mA for 30 s using a Leica EM ACE200 vacuum coater (Leica Microsystems, Wetzlar, Germany). Following, 3 µL of EV isolate was deposited on a grid, which was subsequently blotted (3 s, force: 5) and frozen in liquid ethane using a Vitrobot Mark IV vitrification system (Thermo Fisher, Waltham, MA, USA). Samples were visualized using two levels of magnification on a Tecnai G2 20 TWIN transmission electron microscope (FEI, Hillsboro, OR, USA) equipped with a High-Sensitive 4k × 4k Eagle camera (FEI, Hillsboro, OR, USA).

### 4.7. Electrophoretic Light Scattering (ELS)

ELS measurements were performed with Zetasizer Nano ZS (Malvern Instruments, Worcestershire, UK). U-curved disposable cuvettes (DTS1070; Malvern Instruments, Worcestershire, UK) were used. The viscosity of the dispersant was 0.8872 cP (mPa s), and material absorption was 0.001. The refractive index of the material and water were set to 1.45 and 1.33, respectively. The samples were diluted 20-fold with water before the measurements. The Hückel model was used for zeta potential calculations. Each sample was analyzed in triplicate.

### 4.8. Proteomic Analysis

Three independently obtained isolates (15 µg total protein) were reduced in 5 mM tris(2-carboxyethyl)phosphine at 60 °C for 30 min., alkylated by 40 mM iodoacetamide for 30 min at room temperature, and finally digested using trypsin (Promega, Gold Trypsin, Mass Spectrometry Grade, Madison, WI, USA) applying the sp3 method on amine magnetic beads 20 µg µL^−1^ (MagReSyn, ReSyn Biosciences, Pretoria, South Africa) in KingFisher Flex instrument (Thermo Scientific, Waltham, MA, USA) at 50 °C for 4 h. Then samples were acidified by the addition of 10% formic acid and were analyzed on a timsTOF Pro 2 instrument (Bruker, Billerica, MA, USA) performing data-dependent acquisition (DDA) proteomics, operating in positive-ion mode. Peptides obtained by digestion were separated on a Ultimate 3000 RSLC nano system (Thermo Scientific) with bioZen C18 nano column 250 × 0.075 mm (Phenomenex, Torrance, CA, USA) with trap cartridge Acclaim PepMap C18 5 × 0.3 mm (Thermo Scientific, Waltham, MA, USA) using 30 min gradient from 5% to 35% of acetonitrile with 0.1% (*v*/*v*) formic acid. The mass spectrometer was equipped with an ion mobility separation feature, allowing the analysis of ions in a 1/K0 range of 0.6 to 1.6. Data were acquired over an *m*/*z* range of 100 to 1700, and the collected data were processed for protein identification and quantification. The fragmentation for DDA analysis was performed using the parallel accumulation and serial fragmentation method with standard settings. The acquired mass spectra were analyzed using the Bruker Data Analysis 6.1 software (Bruker Daltonics, Billerica, MA, USA) for mgf files generation, and proteins were identified using the Mascot Server 3.1 (Matrix Science, Boston, MA, USA) against the NCBI sequence database containing proteins for *Mangifera indica* (42,548 protein groups). Search parameters were set as follows: modification—carbamidomethyl C (fixed) or oxidation M (variable); up to 2 missed cleavages; mass tolerance: 25 ppm for precursor mass and 25 ppm for fragment mass; target FDR: 1%.

## 5. Conclusions

In this study, we developed and optimized a dedicated method for the isolation of small EVs from mango fruits. A key advancement of this work was the implementation of a SEC column designed and optimized in our laboratory, which provided the desirable purity of the vesicle preparations. This purification step efficiently removed pectin and other polysaccharide contaminants that typically interfere with EV isolation from plant matrices, enabling the recovery of well-defined vesicular structures suitable for downstream characterization.

Proteomic analysis revealed a diverse range of proteins, indicating potential biological activity of mango-derived EVs. While the function of many of the identified proteins is consistent with the properties traditionally attributed to mango, it might be assumed that some of these health-promoting properties are related to the presence of vesicles. However, it should be emphasized that the antioxidative, antimicrobial, and immunomodulatory properties of mango are also linked with their metabolomic composition, which indicates a complementary mechanism of action. Further investigations should aim to expand beyond proteomic profiling to include metabolomic and transcriptomic analyses, thereby identifying other potential cargo molecules. Moreover, functional studies using mammalian cell models or in vivo systems will be essential to elucidate the biological relevance of these vesicles and to explore their potential applications in nutrition, medicine, and biotechnology.

## Figures and Tables

**Figure 1 ijms-26-11375-f001:**
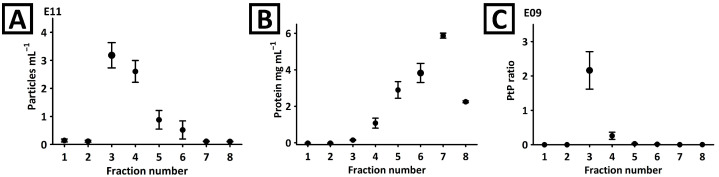
(**A**) NTA and (**B**) BCA analyses of fractions obtained during the separation of mango fruit juice with Sepharose CL-6B column. (**C**) The particle-to-protein (PtP) ratio was determined in each fraction.

**Figure 2 ijms-26-11375-f002:**
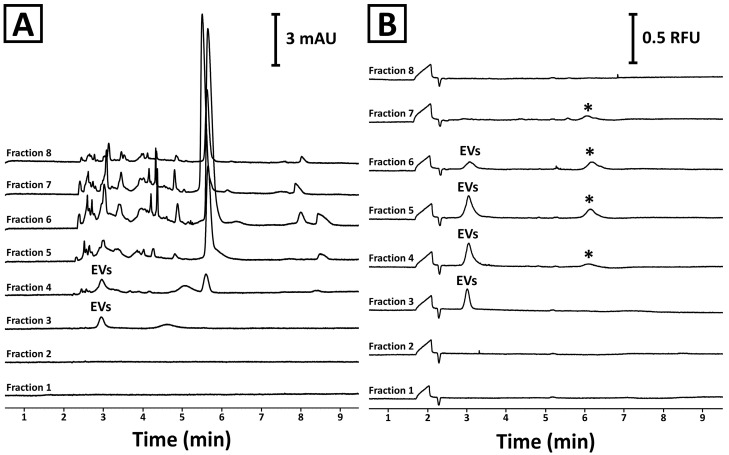
CE analysis of SEC fractions obtained during the separation of mango fruit juice using a Sepharose CL-6B column. (**A**) UV and (**B**) LIF detectors were used in the experiments. The isolation procedure and the analysis conditions are described in Section 4.2 and Section 4.5, respectively. The asterisk (*) indicates an unidentified signal.

**Figure 3 ijms-26-11375-f003:**
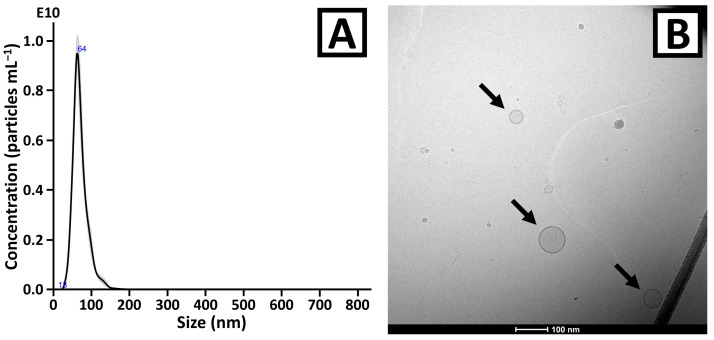
(**A**) NTA and (**B**) cryo-TEM analyses results of the EV isolates obtained from mango fruit juice. In Figure 3A, the trace represents the average from five measurements. The deviation from the average (standard error) was marked with grey. Blue numbers in Figure 3A indicate selected extremes. The vesicles are indicated with arrows in Figure 3B. The zoom-out image of Figure 3B is shown in Appendix A.

## Data Availability

The original contributions presented in this study are included in the article/Appendix A. Further inquiries can be directed to the corresponding author.

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
