# Peer review of "Isolation and Characterization of Extracellular Vesicles Derived from Mango Fruits"

_ijms, 2025, doi:10.3390/ijms262311375_

Round 1

Reviewer 1 Report

Comments and Suggestions for Authors

This study developed a method for isolating extracellular vesicles (EVs) from mango fruit juice and conducted systematic characterization, which holds significant importance for advancing research and utilization of mango-derived EVs. However, certain aspects require improvement, as detailed below:

  1. The mango fruit serves as a critical material in this study. Considering that the protein profile of EVs is likely to vary across different mango cultivars, it is essential that the authors provide detailed information regarding the specific variety (or varieties) used in the experiments.
  2. The study reports the proteomic analysis of mango EVs derived from a single biological sample. This approach raises concerns about the robustness and generalizability of the findings. Furthermore, the authors cite that plant EVs are involved in stress responses (lines 48-49). It is therefore plausible that the proteomic profile of the isolated EVs could be significantly influenced by pre-harvest conditions (e.g., pathogen exposure, nutrient status, drought) or post-harvest handling of the fruits. To strengthen the study, the authors should either:
    a) Perform additional replicates: Isolate and analyze EVs from multiple, independent biological samples to confirm the consistency of the proteomic profile.
    b) Expand the discussion: At a minimum, explicitly discuss the limitation of having only one biological replicate and acknowledge how potential unknown stress conditions might have influenced the identified protein composition, framing it as an important consideration for future research.

Reviewer 2 Report

Comments and Suggestions for Authors

This manuscript describes an SEC-based workflow to isolate small extracellular vesicles (EVs) from mango pulp/juice and presents NTA, cryo-TEM, zeta potential and a DDA proteomic dataset (1,084 proteins). The work is potentially interesting (first proteome for mango EVs), but the current version still need to perform different experiments, as it it over-interprets proteomics alone to claim biological activities. Major additional  experiments to determine their biological effects /use as vehicle of drugs are needed before publication.

Major concerns: 

The article is very descriptive. Please, provide more experiments to describe the potential bioogical effects of mango EVs. 

Please describe why did you choose mango

Why did you use sepharose CL6B? 

It would be of interest to test residuals of pectins/cell wall polysaccharides and the authors state gelation problems with ultrafiltration. No analytic assay (e.g., carbohydrate staining in fractions, monosaccharide analysis, or enzymatic pectinase digestion controls) was provided to quantify residual pectin in the EV fraction. Because pectin can form nanoparticles and entrap proteins, it is essential to show the SEC fraction is not a pectin-rich artefact. Simple carbohydrate assays (e.g., phenol–sulfuric acid), monosaccharide profiling, or pectinase digestion demonstrating that the SEC fraction is not dominated by pectin. Show effect of pectinase on NTA, PtP and CE profiles. Indeed, for me is very weird that EVs appear in this fractions (is too early).

Protein presence alone—without evidence of enzymatic activity in the isolated vesicles, detection of small-molecule cargo (mangiferin, polyphenols), or functional assays—cannot support claims that mango EVs mediate fruit bioactivity in consumers. The Discussion repeatedly links presence → function; this should be softened unless supported experimentally.

please, in addition, incorporate a paragraph discussing the potential gastrointestinal resistance of these vesicles and compare with other plant EVs (ej. https://doi.org/10.1016/j.foodres.2025.117325)

Many plant EV studies include small RNA sequencing and targeted metabolomics (e.g., polyphenols) because those cargoes are key to functional claims. Absence of any RNA/lipid/metabolite data is a major gap for all statements about nutraceutical potential.

If authors want to claim antioxidant/antimicrobial/immunomodulatory activity they must present functional assays (cellular ROS assay, cytokine readouts, antimicrobial MIC) with proper negative controls: vesicle-depleted fractions, heat-inactivated or detergent-disrupted EVs, dye-only controls if using fluorescent uptake. MISEV also suggests dose-response and multiple models.

Please, the evidence of the polyphenol content in EVs is discussed (10.1016/j.phrs.2023.106999) 

Round 2

Reviewer 2 Report

Comments and Suggestions for Authors

Thanks for your carefully revision of my comments. Under my point of view, the paper is suitable for publication.